# Auto/Paracrine C-Type Natriuretic Peptide/Cyclic GMP Signaling Prevents Endothelial Dysfunction

**DOI:** 10.3390/ijms25147800

**Published:** 2024-07-16

**Authors:** Franziska Werner, Takashi Naruke, Lydia Sülzenbrück, Sarah Schäfer, Melanie Rösch, Katharina Völker, Lisa Krebes, Marco Abeßer, Dorothe Möllmann, Hideo A. Baba, Frank Schweda, Alma Zernecke, Michaela Kuhn

**Affiliations:** 1Institute of Physiology, University Würzburg, 97070 Würzburg, Germany; franziska.werner@uni-wuerzburg.de (F.W.); takanalu@kitasato-u.ac.jp (T.N.); lydia_schroeder@web.de (L.S.); katharina.voelker@uni-wuerzburg.de (K.V.); lisa.krebes@uni-wuerzburg.de (L.K.);; 2Institute of Experimental Biomedicine, University Hospital Würzburg, 97080 Würzburg, Germany; schaefer_s7@ukw.de (S.S.); roesch_m4@ukw.de (M.R.); zernecke_a@ukw.de (A.Z.); 3Institute of Pathology, University Hospital Essen, 45147 Essen, Germany; dorothe.moellmann@uk-essen.de (D.M.); hideo.baba@uk-essen.de (H.A.B.); 4Institute of Physiology, University of Regensburg, 93053 Regensburg, Germany; frank.schweda@klinik.uni-regensburg.de

**Keywords:** endothelial dysfunction, arterial stiffening, systolic hypertension, atherosclerosis, angiogenesis, C-type natriuretic peptide, cyclic GMP

## Abstract

Endothelial dysfunction is cause and consequence of cardiovascular diseases. The endothelial hormone C-type natriuretic peptide (CNP) regulates vascular tone and the vascular barrier. Its cGMP-synthesizing guanylyl cyclase-B (GC-B) receptor is expressed in endothelial cells themselves. To characterize the role of endothelial CNP/cGMP signaling, we studied mice with endothelial-selective GC-B deletion. Endothelial EC GC-B KO mice had thicker, stiffer aortae and isolated systolic hypertension. This was associated with increased proinflammatory E-selectin and VCAM-1 expression and impaired nitric oxide bioavailability. Atherosclerosis susceptibility was evaluated in such KO and control littermates on *Ldlr* (low-density lipoprotein receptor)-deficient background fed a Western diet for 10 weeks. Notably, the plaque areas and heights within the aortic roots were markedly increased in the double EC GC-B/*Ldlr* KO mice. This was accompanied by enhanced macrophage infiltration and greater necrotic cores, indicating unstable plaques. Finally, we found that EC GC-B KO mice had diminished vascular regeneration after critical hind-limb ischemia. Remarkably, all these genotype-dependent changes were only observed in female and not in male mice. Auto/paracrine endothelial CNP/GC-B/cGMP signaling protects from arterial stiffness, systolic hypertension, and atherosclerosis and improves reparative angiogenesis. Interestingly, our data indicate a sex disparity in the connection of diminished CNP/GC-B activity to endothelial dysfunction.

## 1. Introduction

Arterial stiffening is an independent and powerful predictor of cardiovascular risk and underlies the development of isolated systolic hypertension [1]. Thickening of the tunica media, with increased collagen and decreased elastin content, enhances vessel rigidity. The reduction in elasticity and distensibility of the arterial wall leads to increased systolic and eventually decreased diastolic blood pressure (BP). Moreover, such structural and functional changes predispose to atherosclerosis [1]. Pulse wave velocity (PWV) is the best method to assess arterial stiffness, aortic PWV being an independent predictor of cardiovascular mortality [2].

Stiffening of large artery walls is both a cause and consequence of arterial hypertension, forming an insidious worsening positive feedback loop [3]. It occurs during ageing and is promoted by cardiovascular risk factors such as obesity and type 2 diabetes mellitus [1,3]. In all these conditions, stiffness correlates with endothelial dysfunction, characterized by reduced nitric oxide (NO) bioavailability and attenuation of the anti-inflammatory and regenerative properties of endothelial cells (Ecs) [4]. Alterations of systemic and local pathways contribute to these pathophysiological changes, including activation of the renin–angiotensin–aldosterone system [5]. The molecular mechanisms cross-linking endothelial dysfunction and vascular stiffening are not completely understood.

Under physiological conditions, the homeostatic cyclic GMP (cGMP)-mediated paracrine vascular actions of NO are complemented and augmented by the endothelial hormone C-type natriuretic peptide (CNP), which also signals through cGMP [6]. CNP, constitutively released by endothelial cells, activates its specific transmembrane guanylyl cyclase receptor, GC-B (gene name *Npr2*), in adjacent smooth muscle cells, pericytes, fibroblasts, and immune cells such as resident perivascular mast cells [6,7,8]. Thereby, CNP diminishes vascular tone and peripheral resistance and contributes to the preservation of the vascular barrier. Mice with endothelial disruption of CNP have mild chronic arterial hypertension and enhanced risk of atherosclerosis and aortic aneurysms, which emphasizes the physiological relevance of paracrine vascular CNP signaling [9,10].

Experiments with exogenous, synthetic CNP have indicated that the hormone also activates GC-B/cGMP signaling in endothelial cells themselves. In vitro, treatment of cultured endothelial cells with CNP elevated their intracellular cGMP levels, stimulated proliferation and vitality, prevented the induction of the vasoconstrictor endothelin (ET-1) and of proinflammatory adhesion proteins, and enhanced the synthesis and release of NO [11,12]. In line with these observations in vitro, the infusion of stabilized CNP to mice augmented post-ischemic vascular regeneration [11]. Moreover, chronic hypertension of mice with endothelial CNP deletion was associated with endothelial barrier dysfunction [9]. The human relevance of such experimental observations is indicated by clinical studies reporting an inverse relationship between CNP plasma levels and arterial alterations such as stiffness and early atherosclerosis [13]. This suggests that endothelial dysfunction is associated with impaired release of not only NO but also CNP. In a reciprocal way, diminished CNP bioavailability could contribute to the progression of endothelial dysfunction and arterial stiffening as part of a “worsening feedback loop”.

To study whether auto/paracrine endothelial CNP/GC-B/cGMP signaling contributes to the preservation of endothelial functions, here we studied mice with EC-specific deletion of the GC-B receptor (EC GC-B KO mice). The results reveal an unexpected sex disparity, with disruption of endothelial GC-B/cGMP signaling leading to diminished NO bioavailability, aortic stiffness, isolated systolic hypertension, enhanced risk of atherosclerosis, and impaired post-ischemic reparative angiogenesis in female but not in male KO mice.

## 2. Results

### 2.1. Female Mice with Endothelial GC-B Deletion Have Isolated Systolic Hypertension

To characterize the physiological role of auto/paracrine endothelial CNP/GC-B/cGMP signaling, we generated EC GC-B KO mice by crossing mice carrying a floxed GC-B gene (GC-B^flox/flox^; gene name *Npr2^fl/fl^*) with transgenic mice expressing Cre recombinase under the control of the endothelial *Tie2* promotor/enhancer [7]. We have previously shown that GC-B is ablated in microvascular Ecs and preserved in vascular smooth muscle cells (VSMCs) and pericytes of such KO mice [7]. To prove the GC-B deletion in macrovascular endothelia, here we compared GC-B mRNA expression in freshly isolated aortic Ecs and the denuded “remaining” aortae between genotypes. qRT-PCR showed that GC-B expression in aortic Ecs from EC GC-B KO mice was almost abolished (Figure 1). In the rest of the aortic wall (containing the tunica media and adventitia), the GC-B expression levels were about 10 times higher than in Ecs and not different between KO and controls (Figure 1).

Our previous studies in EC GC-B KO mice had shown that the vasodilatating effects of CNP in the microcirculation are endothelium-independent and mediated by GC-B activation in distal arteriolar SMCs and pericytes [7]. Here, to address the role of endothelial GC-B signaling overall in the regulation of BP, male and female EC GC-B KO and control littermates were subjected to tail cuff and telemetry recordings. For males, the systolic and diastolic BP levels as well as heart rate did not differ between the two genotypes (Figure 2A). However, female EC GC-B KO mice had significant increases in systolic and mean BP levels by ≈11 mmHg, without changes in diastolic BP or heart rate (Figure 2B). Telemetry confirmed that female KO mice had mildly but consistently elevated systolic BP during the light phase, whereas their diastolic BP, heart rate, and spontaneous locomotor activity were not different from sex-matched control littermates (Figure 2C). Moreover, in accordance with the augmented cardiac pressure load, the female EC GC-B KO mice had enlarged left ventricular (LV) cardiomyocyte cross-sectional areas, with preserved contractile functions (Appendix A).

### 2.2. Female Mice with Endothelial GC-B Deletion Have Stiffer Aortae

To study whether the isolated systolic hypertension of female KO mice was associated with arterial stiffening, we measured aortic pulse wave velocity (PWV). Indeed, female KO mice had significantly higher aortic PWV than control littermates (Figure 3A, right panel). As shown, such genotype-dependent differences were not observed in male mice (left panel).

Aortic stiffness of female KO mice was associated with structural changes: the total, adventitial, and tunica media thickness of the aortic wall were increased (Figure 3B). Immunohistochemistry revealed a rise in collagen content (Figure 3C). The elastin content was unaltered, but there was a significant increase in the number of elastin breaks (Figure 3C).

Taken together, these results reveal that impaired endothelial CNP/GC-B/cGMP signaling promotes arterial stiffening and systolic hypertension in female but not male mice.

### 2.3. Increased Expression of Proinflammatory Endothelial Adhesion Molecules in Aortae from Female EC GC-B KO Mice

In cultured endothelial cells, CNP decreased the expression levels of ET-1 and of proinflammatory adhesion proteins [12]. Patients with vascular stiffness have reduced plasma levels of CNP [13] but increased plasma levels of vascular cell adhesion molecule-1 (VCAM-1) and E-selectin [14]. Therefrom, we hypothesized that increased expression of these endothelial proteins contributes to aortic stiffening in EC GC-B KO females. As shown in Figure 4A, the aortic mRNA expression and plasma levels of ET-1 were similar in KO and control females. The aortic levels of P-selectin also did not differ between genotypes (Figure 4B). However, the aortic levels of E-selectin and VCAM-1 were markedly and significantly increased in EC GC-B KO females (Figure 4B).

### 2.4. Decreased Phosphorylation and Activity of Endothelial NO Synthase (eNOS) in Aortae of EC GC-B KO Female Mice

Biochemical assays with recombinant proteins have shown that cGMP-dependent protein kinase (cGKI) phosphorylates eNOS at Serine_1177_, thereby enhancing calcium-independent NO production [15]. We postulated that *endogenous* endothelial CNP, via GC-B/cGMP/cGKI signaling, enhances such activatory eNOS phosphorylation and, conversely, that EC GC-B KO females therefore have reduced aortic eNOS phosphorylation and activity. Indeed, the levels of Serine_1177_-phosphorylated eNOS were significantly reduced in aortae from KO as compared to control females (Figure 4C). To study whether this is associated with diminished NO formation, we assessed the phosphorylation of vasodilator-stimulated phosphoprotein (VASP) at Serine_239_. The levels of VASP-P-Ser_239_ in vascular SMCs and in platelets have been reported to correlate with NO formation and NO/cGMP/cGKI activity [16]. Notably, the levels of Serine_239_-phosphorylated VASP were significantly reduced in both aortae (Figure 4C) and platelets from EC GC-B KO females as compared to control littermates (Figure 4D). In contrast, the phosphorylation of platelet VASP at Serine_157_ (the site targeted by cAMP-dependent protein kinase, PKA) was not different between genotypes (Figure 4D).

Taken together, these observations indicate that impaired auto/paracrine endothelial CNP/GC-B/cGMP signaling in female EC GC-B KO mice leads to increased expression of proinflammatory E-selectin and VCAM-1 and diminished activity of eNOS.

To follow up the observed sex disparity, we also compared eNOS/VASP signaling in aortae from male KO and control mice. As shown in Appendix A, in aortae from male EC GC-B KO mice the levels of total eNOS were increased, whereas the levels of phosphorylated eNOS were unaltered, in comparison to control littermates. Interestingly, the levels of Serine_239_-phosphorylated VASP were also significantly increased, suggesting the compensatory increase of vascular NO/cGMP/cGKI signaling in EC GC-B KO males (Appendix A).

### 2.5. Female EC GC-B KO Mice Have Enhanced Susceptibility to Atherosclerosis

Arterial stiffening predisposes to atherosclerosis [1,2]. Moreover, mice with endothelial CNP disruption are prone to atherosclerosis, demonstrating the protective role of CNP [9]. To dissect whether the endothelial actions of the hormone participate in this protective effect, we compared the extent of atherosclerosis between EC GC-B KO and control littermates on *Ldlr* (low-density lipoprotein receptor)-deficient background after feeding a Western diet for 10 weeks.

First, we compared CNP mRNA expression levels in different aortic segments. Under normal dietary conditions, CNP expression was slightly greater in the aortic root and arch as compared to the thoracic and abdominal aorta (Figure 5A). Notably, feeding a Western diet significantly increased CNP expression levels in the aortic root, but not in the other segments (Figure 5A).

A Western diet provoked marked atherosclerosis in male and female control and EC GC-B KO mice (all *Ldlr^−/−^*). Oil red stainings of lipid depositions showed that the size and density of the plaques were highest in the aortic arch as compared to the thoracic and abdominal segments, which is in accordance with many published studies [17]. Overall, these changes were greater in females (Figure 5B) than in males (Appendix A), but without differences between genotypes (meaning control and EC GC-B KO mice of the same sex).

Because the Western diet selectively increased CNP expression in aortic roots, we characterized the plaques in this region more carefully. Interestingly, aldehyde-fuchsin stainings showed that the areas and heights of the plaques in the aortic roots were significantly greater in EC GC-B KO as compared to control females (Figure 5C). Concordantly, immunohistochemistry showed increased macrophage infiltration (*p* = 0.06; Figure 6A). Moreover, the number and area of the necrotic cores within these aortic root plaques were greater in KO than in control females (Figure 6B). Again, such genotype-dependent changes were not observed in aortic roots of male mice (Appendix A). The CD31-positive endothelial-covered plaque areas were not different between the two genotypes (Figure 6C).

These results show that impaired endothelial CNP/GC-B/cGMP signaling augments the susceptibility for atherosclerosis in aortic roots of female mice. This is associated with plaque characteristics that are linked to destabilization, a key trigger of the thrombotic complications of atherosclerosis [18].

### 2.6. Female EC GC-B KO Mice Have Diminished Angiogenesis after Ischemia

Endothelial dysfunction is associated with impaired capacity of endothelial regeneration and vascular repair [1,2,3,4]. Therefore, in a final experimental series we studied whether EC GC-B KO mice present diminished angiogenesis.

Confirming published data [11], in primary cultured human endothelial cells (HUVECs) CNP enhanced intracellular cGMP levels and proliferation (BrdU incorporation) in a concentration-dependent way (Figure 7A,B). The effect of CNP on HUVEC proliferation was prevented by Rp-8-Br-PET-cGMPs (100 μM, 15 min pretreatment), an inhibitor of cGKI, indicating the involvement of GC-B/cGMP/cGKI signaling (Figure 7B).

To address the relevance in vivo, we studied vascular regeneration after hind-limb ischemia experimentally generated in mice by femoral artery excision [19]. Serial blood flow measurements by laser Doppler perfusion imaging (LDPI) revealed that the recovery of limb perfusion did not differ between male control and EC GC-B KO mice (Figure 7C, left panel). However, in the EC GC-B KO females, the post-ischemic reperfusion during the second and fourth week after surgery (which is the phase of angiogenesis) was significantly impaired (Figure 7C, right panel). Isolectin stainings of muscle sections demonstrated that the capillary density in the ischemic gastrocnemius muscle was significantly lower in EC GC-B KO as compared with control females (Figure 7D). Accordingly, immunoblots showed diminished expression levels of the endothelial marker protein CD31 (Figure 7E).

Together, these results demonstrate that dysfunctional endothelial CNP/GC-B/cGMP signaling impairs post-ischemic angiogenesis in female mice.

## 3. Discussion

Arterial stiffening predisposes to the development of isolated systolic hypertension and atherosclerosis, and, overall, enhances cardiovascular risk [1,2,3,4]. Endothelial dysfunction not only correlates but also co-causally contributes to the stiffening of large vessels [4]. Here, using a genetic approach in mice, we show that the endothelial hormone CNP, via auto/paracrine activation of its GC-B receptor and cGMP signaling in ECs, improves endothelial functions. Thereby, CNP protects from arterial stiffness, systolic hypertension, and atherosclerosis. Moreover, CNP improves endothelial regeneration and thereby fosters post-ischemic angiogenesis. Conversely, inhibition (KO) of CNP/GC-B/cGMP signaling in endothelial cells provoked endothelial dysfunction, with mild thickening and stiffening of the aortic wall, increased systolic BP, enhanced atherosclerosis in aortic roots, and impaired post-ischemic angiogenesis. Mechanistically, this was associated with enhanced aortic expression of pro-inflammatory endothelial adhesion proteins VCAM-1 and E-selectin, reduced phosphorylation and activity of eNOS, and augmented macrophage infiltration of atherosclerotic plaques in the aortic root. Since the endothelial area covering the plaques apparently was not different between KO and control mice, the reduced anti-inflammatory properties of the GC-B-deficient endothelia may have contributed to such plaque alterations. In line with our observation of a prominent role of CNP in the proximal regions of the aorta, mice with heterozygous or homozygous global GC-B deletion have aortic valve disease and ascending aortic dilatation [20].

### 3.1. The Protective Endothelial Actions of CNP Are Mediated by Two Distinct Receptors

Notably, there were conflicting reports about the receptor and signaling pathways mediating the vascular effects of CNP. Besides activating GC-B, CNP binds to a second receptor “type C” (NPR-C), lacking the guanylyl cyclase domain [6,21]. Both receptors, GC-B and NPR-C, are co-expressed in many types of cells, including endothelia. Mice with global NPR-C deletion exhibit mild arterial hypotension, indicating that NPR-C is a clearance receptor, which captures and internalizes CNP and the cardiac natriuretic peptide ANP [21]. However, other studies in mice have revealed that NPR-C activates G-protein-dependent signaling and mediates biological activities of natriuretic peptides, especially the vascular actions of CNP [9]. Moreover, in cultured human aortic ECs, CNP inhibited endothelial sodium channel (EnNaC) activity by NPR-C activation [22]. Because the vascular phenotype of mice with endothelial CNP inactivation was reproduced in mice with global NPR-C deletion, but not in mice with systemic GC-B deletion, these authors concluded that the vascular actions of CNP are entirely and exclusively mediated by NPR-C [9]. However, such global GC-B knockout mice have a complex systemic phenotype with severe skeletal dysplasia (dwarfism), epilepsy, and high perinatal mortality [23]. In fact, they are so tiny and fragile that it is impossible to perform meaningful cardiovascular studies. To avoid these critical limitations and dissect the endothelial role of GC-B, here we studied mice with endothelial-restricted GC-B deletion, which have unaltered skeletal growth and survival [7]. As already mentioned, the direct vasodilatating actions of CNP in the microcirculation are also preserved in these mice [7]. Despite this, as shown here, they exhibit mild but significant macrovascular stiffening and systolic hypertension. Our data do not exclude a role for NPR-C, but they correct the published concept that the GC-B/cGMP signaling pathway does not participate in the protective vascular actions of CNP.

Supporting the clinical relevance of our experimental observations, GC-B is expressed in human aortic endothelial cells and its expression is decreased in atherosclerotic plaques as well as in stenotic aortic valves [22,24,25]. Moreover, CNP plasma levels are diminished in patients with arterial stiffness and early atherosclerosis [13]. The here presented experimental results indicate that endothelial CNP/GC-B dysfunction might contribute to disease progression.

### 3.2. The CNP–Nitric Oxide Connection

CNP and NO join forces to locally regulate vascular tone and integrity [7,9,10]. Moreover, the following pharmacological interventions indicated that CNP increases eNOS/NO activity: the vasorelaxing responses of isolated arteries to CNP were diminished by pharmacological eNOS inhibition with L-NAME [9]; in rats, infusion of synthetic CNP or vascular CNP overexpression increased the activity of eNOS [26,27]. Supporting a mediatory role for NO in the vascular actions of endogenous CNP, endothelial cell-specific CNP knockout mice had enhanced acute hypertensive reactions to L-NAME [10]. Our studies of control and EC GC-B KO mice complement these studies, revealing that auto/paracrine CNP, via endothelial GC-B/cGMP signaling, stimulates the activatory phosphorylation of eNOS at Serine_1177_. As index of the resultant local NO activity, we evaluated the phosphorylation state of its substrate VASP at Serine_239_ in aortae and platelets. Because platelets express soluble, cytosolic guanylyl cyclase (sGC, the receptor for NO) but none of the particulate, transmembrane GCs (receptors for natriuretic peptides), endogenous VASP-Ser_239_ phosphorylation in platelets can be utilized to estimate luminal NO release and activity [16]. As shown here, ablation of endothelial CNP/GC-B signaling was associated with reduced VASP-Serine_239_ phosphorylation in aortae and platelets. Impaired vascular NO/cGMP signaling might contribute to aortic stiffness and enhanced atherosclerosis of EC GC-B KO mice.

### 3.3. A Sex Disparity in the Link between Diminished CNP/GC-B Activity and Endothelial Dysfunction

The present and previous studies remark sex differences in the role of CNP in the control of vascular functions and blood pressure [9]. As shown here, only the female EC GC-B KO mice had aortic stiffness, systolic hypertension, augmented atherosclerosis, and diminished vascular regeneration. Moreover, whereas eNOS/NO activity was attenuated in EC GC-B KO females, their male KO littermates had even increased aortic eNOS expression levels and NO/sGC/cGMP activity (indicated by VASP-Ser_239_ phosphorylation). Augmented NO production might compensate for impaired CNP signaling in the male mice. In line with this assumption, previous experimental studies suggested that the local regulation of vascular tone in males is more reliant on endothelial NO than on endothelial CNP [28].

The mechanisms accounting for these sex differences remain an important open question for our future investigations. The female sex hormone estradiol induces CNP and GC-B expressions [29,30] and may even directly activate the GC-B receptor to cGMP signaling [31,32]. Moreover, estrogens could stabilize physical and functional interactions between GC-B and eNOS via the association of Hsp90 [33,34,35].

Sex differences in CNP/GC-B expression and/or signaling might also be relevant for vascular stiffening in humans. At the age of 20–50 years, CNP plasma levels are higher in women than in men [36]. Endothelial CNP/GC-B/cGMP signaling might participate in the protective effect of estradiol on endothelial function and arterial stiffness during reproductive age that is dramatically reversed after menopause [1,2].

### 3.4. Summary and Perspectives

In conclusion, this study unravels the protective role of the CNP-activated GC-B/cGMP signaling pathway in the physiological maintenance of endothelial functions. In clinical trials, the dual neprilysin/Angiotensin II receptor blocker sacubitril–valsartan (Entresto^R^, which stabilizes endogenous ANP and especially CNP levels) was superior to a selective Angiotensin II receptor blocker (olmesartan) in reducing arterial stiffness and systolic hypertension [37]. Moreover, in line with our experimental observations, treatment with sacubitril–valsartan caused greater overall reductions in aortic stiffness and systolic hypertension in women than in men [38,39]. Such sex disparities in the pathophysiology and therapy of arterial stiffness might be partly related to sex- (and age)-dependent differences in endothelial CNP release and/or signaling. The therapeutic development of stabilized designer peptides could overcome age- and gender-dependent differences in CNP bioavailability for vascular, i.e., endothelial, protection [40]. The here described endothelial CNP effects, together with the other anti-inflammatory properties of this peptide, may contribute to its favorable inhibitory actions on tissue remodeling, i.e., in heart and renal fibrosis [41,42].

## 4. Materials and Methods

Please see the Tables in the Appendix A for detailed materials. All protocols and non-commercial materials will be made available to any researcher.

### 4.1. Genetic Mouse Models

Homozygous *GC-B^flox/flox^* mice were crossed with *Tie2-Cre^TG^* mice to generate mice with endothelial deletion of GC-B (EC GC-B KO) [7]. *GC-B^flox/flox^* littermates without Cre^tg^ served as controls within each experiment [7]. For studies of atherosclerosis, such mice were crossed with low-density lipoprotein receptor-deficient mice (*Ldlr^−/−^*; Jax: *B6.129S7-Ldlrtm1Her/J*, Strain #:002207) to generate *GC-B^flox/flox^/Ldlr^−/−^* and EC GC-B KO/*Ldlr*^−/−^ mice. Two-month-old mice were fed a Western type of diet for 10 weeks (21% fat, 0.15% cholesterol, 19.5% casein; Altromin, Germany) [17]. Studies were performed with 4–6-month-old littermate mice with a mixed (C57Bl6/J; 129SV) background. All animal studies complied with the Guide for the Care and Use of Laboratory Animals (National Institutes of Health publication no. 85-23, revised 1996) and were approved by the local animal care committees (Regierung von Unterfranken, approval number 55.2 2532-2-690-14).

### 4.2. Hemodynamic Studies

Arterial blood pressure (BP) was monitored in awake mice by tail cuff plethysmography and radiotelemetry [7]. Terminal closed chest hemodynamic studies of left ventricular (LV) function were performed with a retrogradely inserted Micro-Tip pressure–volume catheter [39]. Aortic pulse wave velocity (PWV) was measured noninvasively with Doppler ultrasound (Indus Mouse Doppler System, Webster, TX, USA). After surgical femoral artery dissection, a laser Doppler perfusion imaging (LDPI) instrument (Moor Instruments) was used to estimate the reperfusion in the ischemic limb relative to that in the nonischemic limb [19]. All these studies were performed in isoflurane (2%)-anaesthetized mice after preemptive analgesia with buprenorphine (0.05–0.1 mg/kg BW s.c.).

### 4.3. Real-Time RT-PCR and ELISA

The mRNA expression levels of CNP, GC-B, P/E-Selectins, VCAM-1, and endothelin-1 (ET-1) in freshly prepared aortic endothelial cells as well as in endothelium-denuded and whole aortae were determined by real-time RT-PCR [7,43]. Ribosomal S12 or β2-microglobulin served for normalization. Details for primers and probes are listed in the Table of the Appendix A. Mouse ET-1 plasma levels were determined by ELISA (R&D Systems, Minneapolis, MN, USA).

### 4.4. Histology, Immunohistochemistry, and Morphometry

Paraffin-embedded LV and gastrocnemius muscle sections were stained with PAS (for determination of cardiomyocyte cross-sectional areas) or isolectin (for capillary densities), respectively [19,41]. Aortic sections were stained with 0.1% picrosirius red (for collagen and arterial wall thickness) and Elastica Van Gieson (for elastic fibers). The extent of atherosclerosis throughout the aorta was assessed with Oil-red-O staining [17]. In addition, serial cryosections (5 µm) of aortic roots were stained with aldehyde fuchsin (basic fuchsin and acid aldehyde; Sigma Aldrich, Deisenhofen, Germany) [17]. Macrophages within the plaques were immunostained with antibodies to Mac2 (Cedarlane Labs) and endothelial cells were immunostained with anti-CD31 antibodies (R&D Systems). Nuclei were counter-stained by 4′, 6-Diamidino-2-phenylindole (DAPI, Vector Laboratories) [17].

### 4.5. Studies with Freshly Isolated and Primary Cultured Endothelial Cells

Mice aortae were cut into 1.0 cm rings, opened, and pressed with the luminal side onto microscope slides, which were then placed onto a copper block previously cooled on dry ice. After freezing, the upper wall of the aorta (media and adventitia) was lifted and transferred to TRIZOL for RNA extraction. The frozen endothelial cell layer attached to the slide was also lysed with TRIZOL. Aortic endothelial cells from five mice and endothelial-denuded aortae from two mice were pooled for RNA isolation. Primary cultures of human umbilical vein endothelial cells (HUVECs) were prepared as described and passaged twice for experiments [19].

For determination of cGMP responses, HUVECs were incubated with the phosphodiesterase inhibitor 3-isobutyl-1-methylxanthine (0.5 mM IBMX, 15 min; Sigma Aldrich), and then with CNP (Bachem, Bubendorf, Switzerland) or vehicle (saline) for an additional 15 min. Intracellular cGMP contents were determined by RIA [19]. Proliferation was assessed by 5-bromo-2-deoxyuridine incorporation in HUVECs treated with CNP or vehicle for 24 h (BrdU incorporation kit from Abcam).

### 4.6. Immunoblotting

Tissue proteins were extracted with lysis buffer (Thermo Fisher Scientific, Karlsruhe, Germany) containing protease and phosphatase inhibitors. SDS page and immunoblotting were performed as described [19,43]. GAPDH was used to control protein loadings. Antibodies are detailed in the Table provided in the Appendix A Section.

### 4.7. Statistical Analysis

Results are presented as mean ± SEM (number of experiments are described in the figure legends). Analyses were performed using GraphPad Prism. Data were tested for normality (Shapiro–Wilk test) and equal variance (F test). For statistical analysis of two groups of normally distributed data, a two-tailed Student’s *t*-test was used; for data that were not normally distributed, the Mann–Whitney U test was used. For multiple-group comparisons of normally distributed data, the two-way ANOVA followed by the multiple-comparison Bonferroni’s *t*-test was used. For data that were not normally distributed, the Kruskal–Wallis analysis was performed. The applied tests are specified in the respective Figure legends.

## Figures and Tables

**Figure 1 ijms-25-07800-f001:**
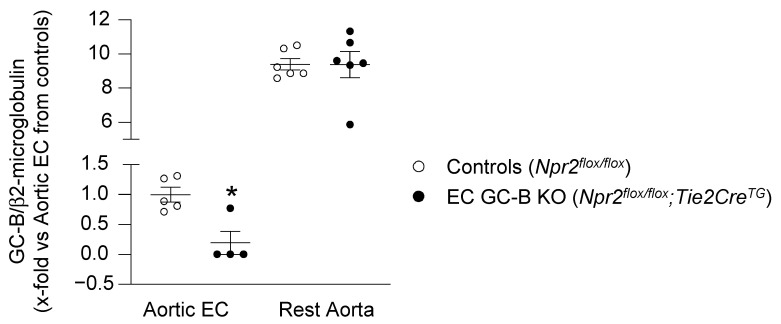
Demonstration of the deletion of the GC-B receptor in aortic endothelial cells. qRT-PCR showed significantly reduced GC-B mRNA expression (normalized to β2-microglobulin) in freshly enriched aortic endothelial cells (Ecs) of EC GC-B KO mice and unaltered GC-B expression in the rest of the aortic wall (*n* = 4–6, * *p* < 0.05 versus controls; Mann–Whitney test).

**Figure 2 ijms-25-07800-f002:**
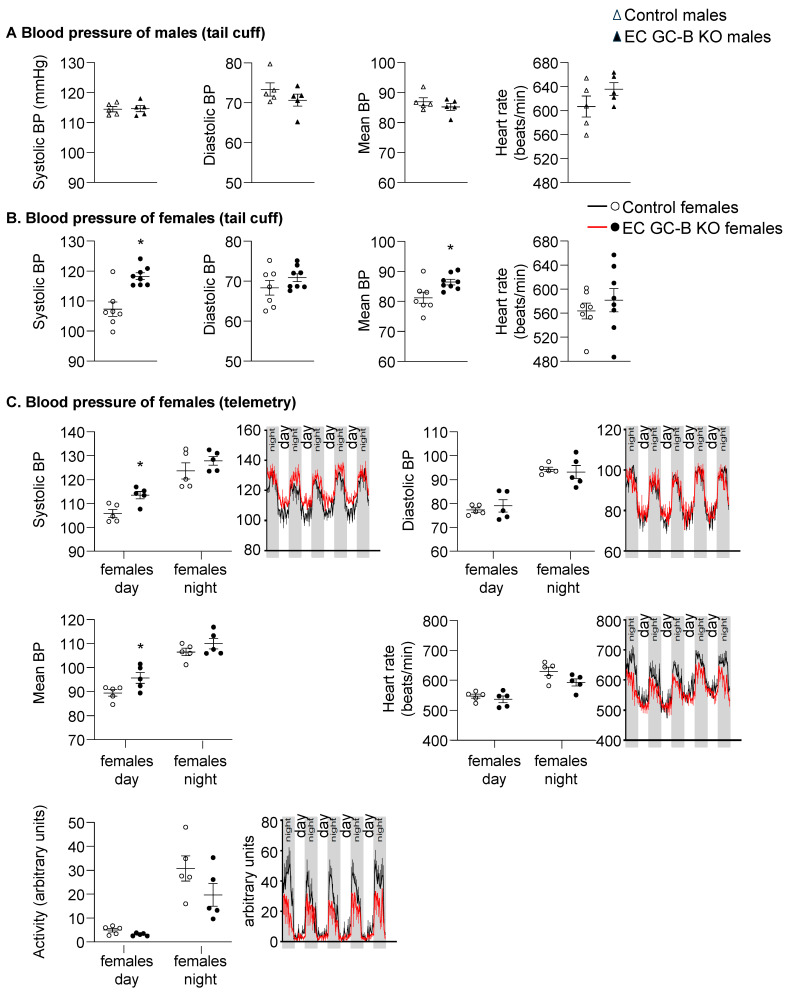
Increased systolic BP in female EC GC-B KO mice. (**A**,**B**) Tail cuff plethysmography showed unaltered systolic and diastolic blood pressure (BP) in male EC GC-B KO mice (top panels) but increased systolic and mean BP in female KO mice as compared to respective sex-matched control littermates (lower panels) (*n* = 5–8). (**C**) Telemetry confirmed increased systolic BP in female KO mice at daytime. Heart rate and locomotor activity did not differ between genotypes (*n* = 5). * *p* < 0.05 versus controls (unpaired *t*-test).

**Figure 3 ijms-25-07800-f003:**
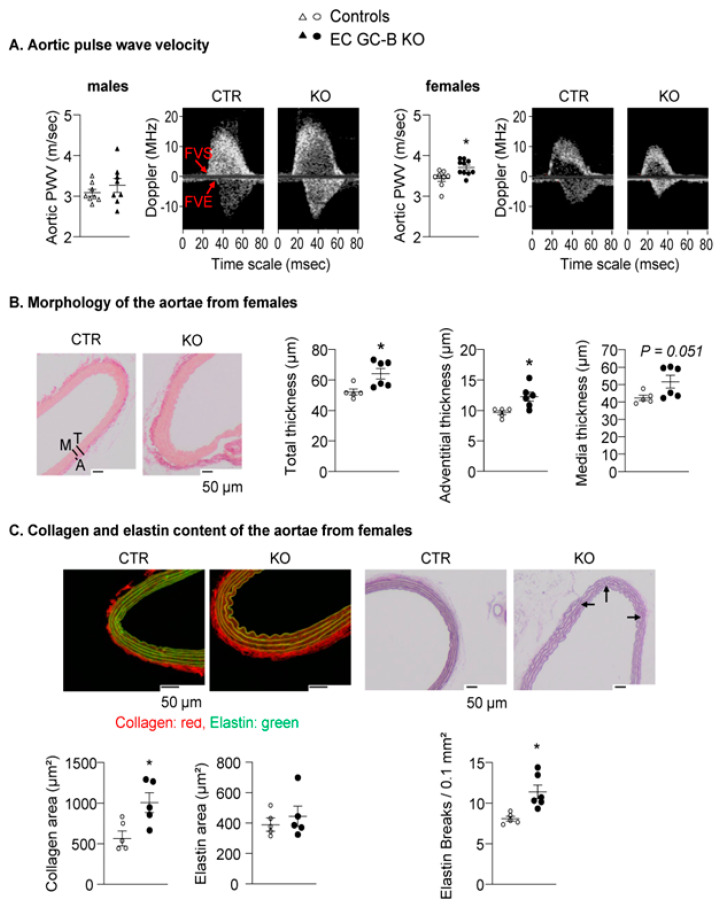
Aortic stiffening and thickening in female EC GC-B KO mice. (**A**) Non-invasive Doppler measurements showed increased aortic pulse wave velocity (PWV) in female, but not male, EC GC-B KO mice as compared to their control littermates (*n* = 8–10). (**B**) Hematoxylin–eosin stainings showed enhanced total, adventitial, and medial thickness of aortae from female EC GC-B KO mice; representative pictures are in right panels. (**C**) Sirius red and Elastica Van Gieson stainings revealed increased collagen area, unchanged elastin area, and increased numbers of elastin breaks (see arrows) in aortae of female EC GC-B KO mice; representative pictures are on top (*n* = 5–6). * *p* < 0.05 versus controls (unpaired *t*-test).

**Figure 4 ijms-25-07800-f004:**
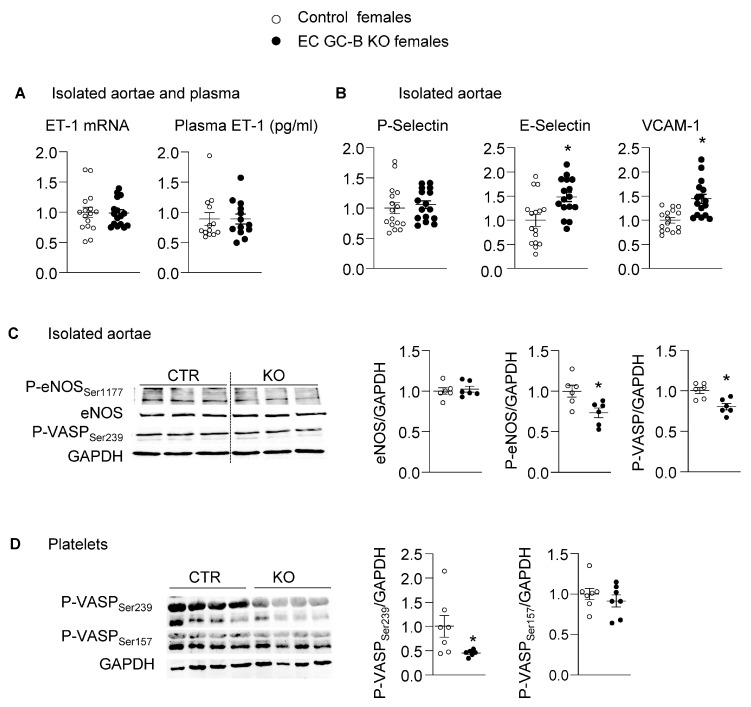
Enhanced expression of proinflammatory adhesion proteins and decreased eNOS phosphorylation and activity in aortae from female EC GC-B KO mice. (**A**) Aortic ET-1 mRNA expression levels (qRT-PCR) and ET-1 plasma concentrations (ELISA) were similar in KO and control females (*n* = 13, unpaired *t*-test). (**B**) Unchanged P-selectin but increased E-selectin and VCAM-1 mRNA expression levels in aortae from KO females. Target mRNAs were normalized to S12 and calculated as x-fold vs. controls (n = 16; * *p* < 0.05 versus controls; unpaired *t*-test). (**C**,**D**) Immunoblots: reduced eNOS-Serine_1177_ and VASP-Serine_239_ phosphorylation in aortae (**C**) and platelets (**D**) from female KO mice; platelet VASP-Serine_157_ phosphorylation was unaltered. Target proteins were normalized to GAPDH and calculated as x-fold vs. controls (n = 6–7; * *p* < 0.05 versus controls; two-way ANOVA).

**Figure 5 ijms-25-07800-f005:**
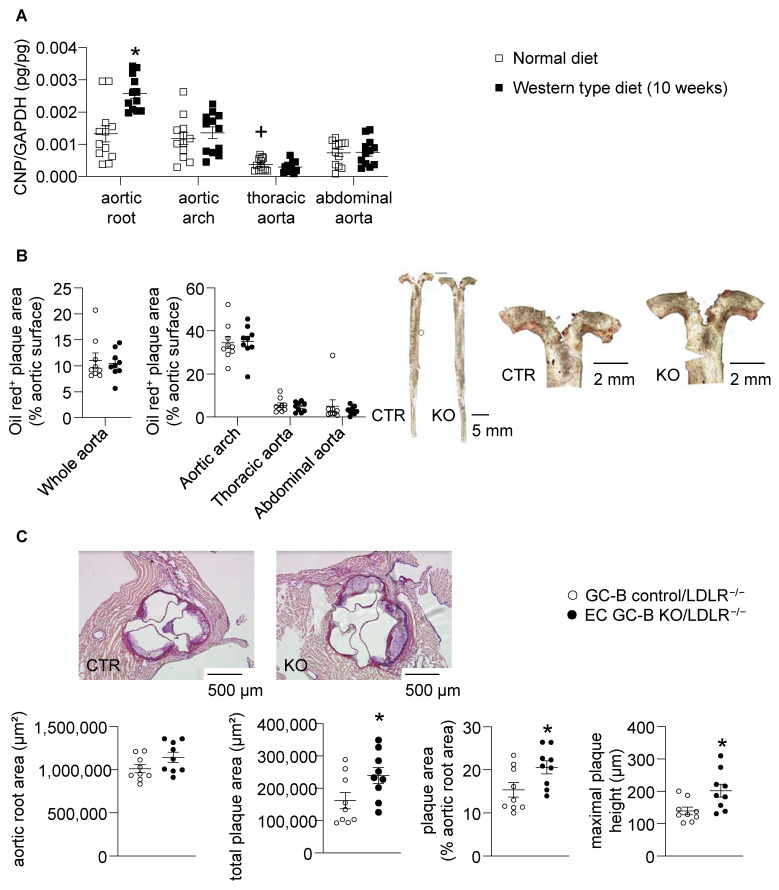
Western diet increased CNP mRNA expression in aortic roots and EC GC-B KO females develop enhanced atherosclerosis in this region. (**A**) Quantitative qRT-PCR: CNP mRNA expression (normalized to S12) was greater in the aortic root as compared to distal segments and increased after Western diet (*n* = 12 samples from 6 mice; *p* < 0.05 vs. ^+^ aortic root, or * normal diet; two-way ANOVA). (**B**) Oil red stainings: plaque areas in the whole aorta and specifically in the aortic arch as well as thoracic and abdominal segments were not different between control and EC GC-B KO females (all with *Ldlr^−/−^* and Western diet; *n* = 9; Mann–Whitney test). Right panels: Representative pictures. (**C**) Aldehyde-fuchsin stainings showed that in the aortic roots of EC GC-B KO/*Ldlr^−/−^* females the total and relative plaque areas as well as plaque heights were increased. Representative pictures are on top (*n* = 9; * *p* < 0.05 vs. control/*Ldlr^−/−^* females; unpaired *t*-test).

**Figure 6 ijms-25-07800-f006:**
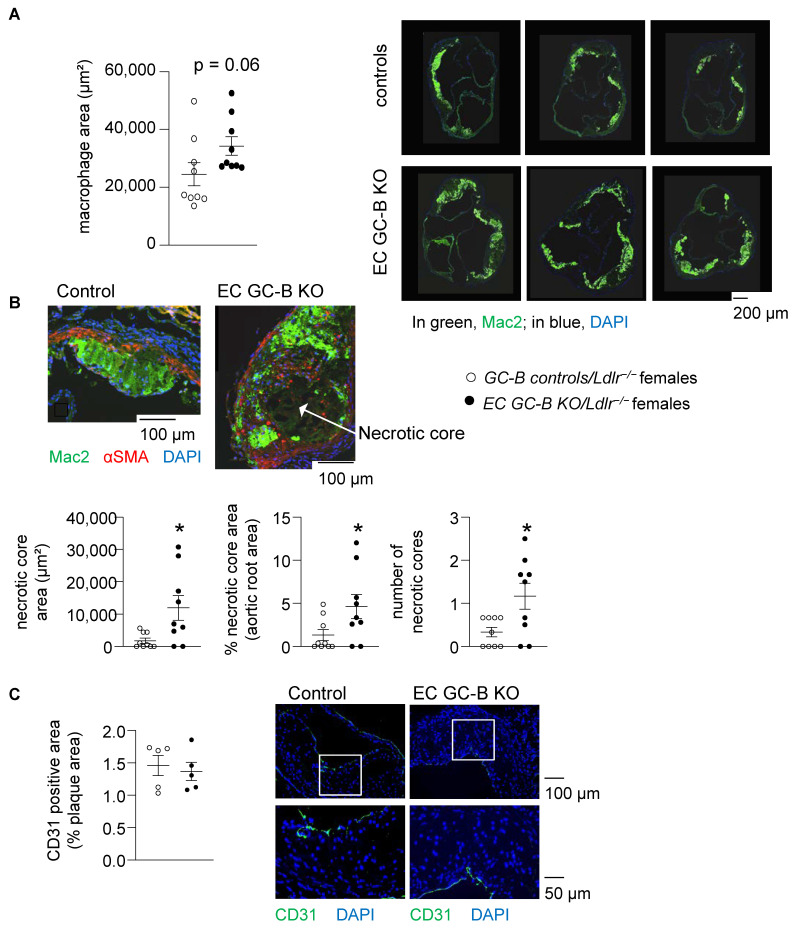
The atherosclerotic plaques in the aortic roots of EC GC-B KO female mice have increased macrophage infiltrations and greater necrotic cores despite unaltered endothelial coverage area. (**A**) Immunocytochemistry: increased macrophage areas in atherosclerotic aortic roots of EC GC-B KO/*Ldlr^−/−^* females; right panels: examples of macrophage (Mac2) stainings (*n* = 9; *p* = 0.06 vs. control/*Ldlr*^−/−^ females; unpaired *t*-test). (**B**) Morphometric analyses of aortic root plaques (stained for macrophages (Mac2) and smooth muscle cells (αSMA)) showed increased areas and numbers of necrotic cores (arrows) in such KO females (*n* = 9, * *p* < 0.05 vs. controls; unpaired *t*-test). (**C**) Stainings of endothelial cells (CD31) did not reveal differences in CD31^+^ areas within aortic root plaques of the two genotypes; exemplary pictures of green-labeled endothelial cells are shown in the right panels (n = 5; Mann–Whitney test). The two lower panels show the tissue areas labeled with white squares in the upper panels, at higher magnification.

**Figure 7 ijms-25-07800-f007:**
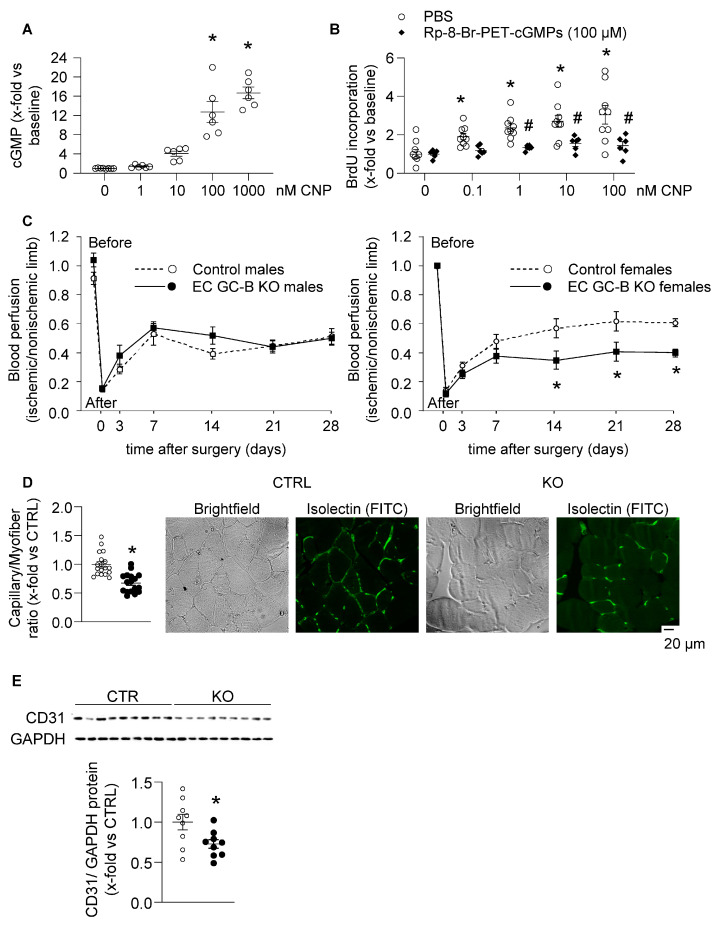
Impaired post-ischemic endothelial regeneration in female EC GC-B KO mice. (**A**) CNP enhanced cGMP levels in HUVECs (*n* = 6–8; * *p* < 0.05 vs. basal (0); one-way ANOVA). (**B**) BrdU incorporation assays: CNP stimulated HUVEC proliferation and this effect was abolished by pretreatment with the PKGI inhibitor Rp-8-Br-PET-cGMPs (100 µM) (*n* = 6–9; *p* < 0.05 vs. * basal (0) or ^#^ PBS; two-way ANOVA). (**C**) Laser Doppler perfusion imaging before/after experimental hind-limb ischemia (HLI): post-ischemic reperfusion was unaltered in male (left panel) but reduced in female EC GC-B KO mice (right panel) in comparison to sex-matched control littermates (*n* = 6–9; * *p* < 0.05 vs. controls; two-way ANOVA). (**D**) Isolectin stainings of gastrocnemius muscles dissected 28 days after HLI revealed reduced capillary/myofiber ratios in female KO mice (*n* = 18 sections from 9 mice; * *p* < 0.05 vs. controls; Mann–Whitney test); right panels: exemplary pictures. (**E**) Immunoblots: Reduced CD31 expression in ischemic gastrocnemius muscles of female KO mice. Levels were normalized to GAPDH and expressed as x-fold vs. controls (*n* = 9; * *p* < 0.05 vs. controls; unpaired *t*-test).

## Data Availability

Data are contained within the article and Appendix A.

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
