# Peer review of "Auto/Paracrine C-Type Natriuretic Peptide/Cyclic GMP Signaling Prevents Endothelial Dysfunction"

_ijms, 2024, doi:10.3390/ijms25147800_

Round 1

Reviewer 1 Report

Comments and Suggestions for Authors

This manuscript is well-organized and contains a lot of data to support the hypothesis. I would like to recommend this manuscript to be publish in its current form. 

This manuscript, titled Auto/Paracrine C-Type Natriuretic Peptide/Cyclic GMP signalling Prevents Endothelial Dysfunction, is very interesting and unique. The authors characterized the role of CNP/cGMP signalling in endothelial dysfunction. To test their hypothesis, they employed an endothelial cell-specific guanylyl cyclase-B knockout (ecGC-B KO) mouse model. They demonstrated that GC-B KO mice developed systolic hypertension that was associated with increased E-selectin and VCAM-1 expression. They further used the double knockout mouse for ecGC-B and low-density lipoprotein receptors and showed that this double KO mouse developed increased atherosclerotic plaques with enhanced macrophage infiltration and greater necrotic cores. Additionally, they showed that vascular regeneration was decreased in the ecGC-B KO mouse after critical hind-limb ischemia. Most interestingly, they found all these pathophysiological and functional changes took place in the female mice only. Overall, the manuscript is well-written and carried out a lot of experiments to support their findings. I have only a few concerns in this manuscript.  
  1. Since this manuscript emphasized the role of CNP/GC-B/cGMP signaling in endothelial dysfunction, performing some functional assays, including the tube formation assay using the primary ECs from GC-B KO mice or the aortic ring sprouting assay would consolidate EC function.
  2. Atherosclerosis is associated with increased microvasculature in atherosclerotic plaques, which increases the likelihood of plaque rupture. However, one of their findings suggests that vascular regeneration was decreased in the ecGC-B KO mouse. Therefore, the authors should address this contradiction. They can perform immunostaining of the atherosclerotic lesion for endothelial cell markers, including CD31 or lectin.

Author Response

Our answers to Reviewer 1:

Many many thanks for your positive evaluation and interest!

  • The aortic ring assay of angiogenesis was previously established in our group and we have used this assay to test the effects of CNP. Unfortunately, in this assay in our hands ANP, but not CNP, exerted proangiogenic effects. Therefore, we did not study the aortae from EC GC-B KO mice in this angiogenesis model. We think that the hind-limb ischemia experiments illustrated in Figure 7 (Figure 7 C-E) present a much better model of arteriogenesis and angiogenesis in vivo. We hope very much that the reviewer will accept these results as they are because we will not be able to establish an additional angiogenesis model within the time frame available for the revision of our present manuscript.
  • As requested, we immunostained the atherosclerotic plaques for the endothelial marker CD31. Morphometrical analyses did not reveal differences between the two genotypes. Please see lines 235-236, Figure 6C, lines 250-259, 310-313 and 446 in the revised manuscript.

Reviewer 2 Report

Comments and Suggestions for Authors

This paper by Werner et al. entitled “Auto/Paracrine C-Type Natriuretic Peptide/Cyclic GMP  Signalling Prevents Endothelial Dysfunction” shows experimental data obtained in mice, supporting a beneficial gender (female)-dependent auto/paracrine endothelial CNP/GC-B/cGMP signaling. As a matter of fact, CNP has so far attracted the attention of researchers to a much lesser extent as compared to that of the other 2 natriuretic peptides ANP and BNP in the field of cardiovascuar and renal dieseases. This interesting paper unraveled the CNP-induced, genotype-dependent protection from arterial stiffening, systolic hypertension and atherosclerosis.     

 Specific Comments

1.     Bridging experimental to human data on CNP, to further strengthen results of this study, please bring  to the fore in the Discussion some previous reports on: A) increased plasma and urinary CNP in nephrotic syndrome and the lowering effect of low protein diet (Cataliotti et al., Am J Phisiol Renal Physiol 2002), because it was a pioneer evidence of the nephro-protective role of CNP, possibly due endothelial response to vascular stress induced by proteinuria. B) In keeping with these findings, further data in rats demonstrated that age-related renal fibrosis was correlated with increased urinary CNP (Sangaralingham et al., Am J Phisiol Renal Physiol 2011), likely counterbalancing renal fibrosis. C) By contrast, plasma CNP, unlike ANP and BNP, failed to correlate to cardiac hypertrophy in end stage renal disease, likely because of the advanced stage of endothelial damage associated with this condition (Cataliotti et al., Mayo Clin Proc 2001). All in all, these data, while running in parallel with animal data of the present study, were also consistent in paving the way to a better  and wider understanding of the role of CNP in the pathphysiology of cardio-reno-vascular damage.

2.      Fig.5, Panel A: symbols showing data on normal and western type diets look similar and should be modified to highlight difference.

Author Response

Our answers to Reviewer 2:

Many many thanks for your positive evaluation and interest!

According to your valuable suggestion we have completed the discussion section with a paragraph mentioning that the endothelial anti-inflammatory effects of CNP may participate in the previously reported protective antifibrotic actions of the peptide. Please see lines 400-403 of the manuscript and corresponding references 41 and 42.

In addition we have revised the symbols of Figure 5, panel A. Unfortunately I was not able to substitute this Figure in the document which I had received from the journal. Therefore I will include the revised Figure 5 (as well as revised Figure 6) as separate files, to be integrated by the journal (please!).